# Matrix prior for data transfer between single cell data types in latent Dirichlet allocation

**Alan Min[1], Timothy Durham[2,3], Louis Gevirtzman[2], William Stafford Noble [2,4] ***

**1** Department of Statistics, University of Washington, Seattle, Washington, United States of America,
**2** Department of Genome Sciences, University of Washington, Seattle, Washington, United States of America,
**3** Broad Institute, Cambridge, Massachusetts, United States of America, **4** Paul G. Allen School of Computer
Science and Engineering, University of Washington, Seattle, Washington, United States of America

* wnoble@uw.edu

## Abstract

Single cell ATAC-seq (scATAC-seq) enables the mapping of regulatory elements in fine-grained cell types. Despite this advance, analysis of the resulting data is challenging, and large scale scATAC-seq data are difficult to obtain and expensive to generate. This motivates a method to leverage information from previously generated large scale scATAC-seq or scRNA-seq data to guide our analysis of new scATAC-seq datasets. We analyze scATAC-seq data using latent Dirichlet allocation (LDA), a Bayesian algorithm that was developed to model text corpora, summarizing documents as mixtures of topics defined based on the words that distinguish the documents. When applied to scATAC-seq, LDA treats cells as documents and their accessible sites as words, identifying "topics" based on the cell type-specific accessible sites in those cells. Previous work used uniform symmetric priors in LDA, but we hypothesized that nonuniform matrix priors generated from LDA models trained on existing data sets may enable improved detection of cell types in new data sets, especially if they have relatively few cells. In this work, we test this hypothesis in scATAC-seq data from whole *C. elegans* nematodes and SHARE-seq data from mouse skin cells. We show that nonsymmetric matrix priors for LDA improve our ability to capture cell type information from small scATAC-seq datasets.

## Author summary

Identifying cell types based on genomics information is an important task but can present challenges because genomics information can be high-dimensional and contain many zeros. Previous work has used latent Dirichlet allocation (LDA), a method that automatically identifies "topics" within a dataset, and has used these topics to better understand the cell types within a population. LDA has been applied to single cell ATAC-seq datasets, which provide information about open chromatin regions within individual cells. We focus on improving the LDA framework by enabling the incorporation of auxiliary forms of information. In particular, we present a method that uses data from large reference populations of cells to aid in the formation of topics for a smaller, target population of cells. We demonstrate first, through simulation, that our method can recover topics when

**Data Availability Statement:** The code to run LDA with a matrix prior is available at https://github.com/gevirl/LDA using the -betaFile option. There are no primary data in the paper. A vignette of

running this software is available at https://github.com/Noble-Lab/lda_matrix_prior.

**Funding:** This material is based upon work supported by the National Science Foundation Graduate Research Fellowship under Grant No. DGE-1762114. The funders had no role in study design, data collection and analysis, decision to publish, or preparation of the manuscript.

**Competing interests:** The authors declare no competing interests.

the data follows the assumptions of our model. We then use a dataset of mouse skin cells and another with *C. elegans* cells to demonstrate that in a real data setting, our method improves the quality of topics recovered from the genomics data.

## Introduction

Single cell genomics has emerged as a powerful method to characterize gene expression (scRNA-seq) and chromatin accessibility (scATAC-seq). The resulting data enables fine-grained identification of cell types. For example, in *Caenorhabditis elegans*, scRNA-seq and scATAC-seq have been used to measure genome-wide gene expression levels and chromatin accessibility for the majority of individual cells in the developing embryo and second-stage larval (L2) worms [1–3].

Several research groups have found that a Bayesian modeling approach called latent Dirichlet allocation (LDA) is an effective method for distinguishing different cell types in scRNA-seq and scATAC-seq data [4, 5]. LDA was developed to model topics in text corpora using counts of words in each document, but when applied to scATAC-seq data, can be used to condense peaks into topics that describe cell types within the data. When applied to scATAC-seq data, the outputs of LDA are a cell-topic matrix, describing the topics assigned to each cell, and a topic-peak matrix, describing how strongly a peak contributes to the definition of each topic. LDA is also well-suited to model single cell genomics data because it expects a matrix of integers as input, and thus can naturally operate on the raw count matrices generated by scATAC-seq or scRNA-seq.

Despite promising results, the challenges posed by scATAC-seq data motivated us to incorporate auxiliary data into the LDA algorithm. Single cell data is still expensive to gather, but there are large compendia of single cell ATAC-seq data available. We aim to use large reference sets of scATAC-seq data ("atlases") to improve the analysis of smaller datasets through the use of LDA with a nonuniform matrix prior. Specifically, we propose to use previously generated data to create a probabilistic prior for use by the scATAC-seq LDA model. We further investigate the possibility of using scRNA-seq data to transfer information to smaller scATAC-seq datasets. In general, the Bayesian prior methodology provides a principled and computationally lightweight way to incorporate auxiliary data.

We verified the utility of our approach via simulation and then applied the technique to a dataset from *C. elegans* produced using the sci-ATAC-seq assay [1] and a dataset from mouse skin cells produced using the SHARE-seq assay [6]. We first used simulated data to verify the feasibility of transferring information between two datasets with the same, known, underlying topics; and we show that the nonuniform matrix prior can increase the ability of LDA to identify true underlying topic structure within a given dataset. Next, for the *C. elegans* and mouse skin scATAC-seq data, we split each full data set into a larger "reference" subset and a smaller "target" subset, then applied LDA with a uniform symmetric prior to the reference subset and used the results of that LDA as a nonuniform matrix prior for an LDA model of the target subset. We report that in the mouse skin data, agreement with previously called cell types improved by using auxiliary scATAC-seq data, and that correlation of the output matrices from the "target" LDA with the output matrices from the LDA on the full data set is higher with the nonuniform matrix prior than with the uniform symmetric prior. For the *C. elegans* data, we also found increased correlation of the output matrices between the full data set LDA and the target LDA when using the matrix prior; however, unlike with the SHARE-seq data, we saw no improvement in the agreement with previously called cell types. Finally, we

leveraged the paired nature of the scATAC-seq and scRNA-seq data in the mouse skin SHARE-seq dataset to attempt to transfer information across single cell assays. We used the output from an LDA on the scRNA-seq data as a matrix prior for an LDA on the scATAC-seq data. In this case, we did not see a clear improvement when using the matrix prior, but the cross-assay matrix prior might still be improved with further hyperparameter tuning.

# 1 Approach

## 1.1 Background: Latent Dirichlet allocation

LDA was originally developed to model text documents as a mixture of topics. For a fixed vocabulary, a document is described as the number of occurrences of each word in the vocabulary. In our case, instead of words, we are modeling scATAC-seq peaks or scRNA-seq genes. We assume that each cell is generated from a mixture of topics, and that each topic has a distribution of peaks or genes from the vocabulary associated with it. The parameters we hope to estimate are the distribution of topics for each cell and the distribution of peaks or genes for each topic, called the "cell-topic matrix" and the "topic-peak matrix" or "topic-gene matrix," respectively. A generative model is assumed for each of the cells, described below, following [7].

Let $N$ be the number of sequencing reads or unique molecular identifiers (UMIs) in a cell. Let $T$ be the number of topics. Let $V$ be the number of peaks or genes in the vocabulary. Let $U$ be the number of cells. Let the vector $\mathbf{w} = (w_1, \ldots, w_V)$ be a cell, where each $w_i$ is the number of reads or UMIs associated with a particular peak or gene from the vocabulary. Let $\boldsymbol{\theta} = (\theta_1, \ldots, \theta_T)$ be the topic distribution of a given cell, where $\sum_{i=1}^T \theta_i = 1$. Let $\boldsymbol{\phi}_t = (\phi_{t1}, \ldots, \phi_{tV})$ be the peak or gene distribution for topic $t$ such that $\sum_{w=1}^V \phi_{tw} = 1$, where $\phi_{tw}$ indicates the probability of observing a read or UMI from the peak or gene $w$ given that its topic was $t$. Let $\boldsymbol{\alpha}$ be the "basis vector" of length $T$ such that $\sum_i \alpha_i = 1$ and $\alpha_i > 0$ for each $i$, which parameterizes the Dirichlet prior distribution over the cell-topic matrix. Similarly, let $\boldsymbol{\beta}$ be a basis vector of length $V$ such that $\sum_j \beta_j = 1$ and $\beta_j > 0$ for each $j$, which parameterizes the Dirichlet distribution over the topic-peak or topic-gene matrix. Let $c_\alpha$ be the "concentration parameter" for $\boldsymbol{\alpha}$, and $c_\beta$ be the concentration parameter for $\boldsymbol{\beta}$. Let $\xi$ be the average number of peaks or genes in a cell. Notation is summarized in S1 Table.

We note that $\boldsymbol{\alpha}$ is a prior that describes the frequency at which topics are observed, and $\boldsymbol{\beta}$ is a prior that describes the frequency at which peaks or genes are observed, i.e. a large value of $\alpha_i$ or $\beta_i$ indicates that the $i$th topic or $i$th peak or gene is more common, respectively.

[7] first describe a variational Bayes approach to optimizing the posterior distribution of the topic assignments for each peak or gene. Following [4] and [8], however, we use a Gibbs sampling algorithm for LDA for scATAC-seq data (Algorithm 1).

**Algorithm 1**: LDA Gibbs sampling algorithm to estimate topic assignments for each word, by calculating $p(z_{ij} = k | \mathbf{z}^{(-i)}, \mathbf{w})$, the probability of topic $z_{ij}$ being $k$ given all other topic assignments and all documents

```
Result: Topic-peak matrix n_kw estimating φ and cell-topic Matrix n_dk
estimating θ
Input: Cells {d_i} each with peaks w_ij
Assign counts n_dk, n_kw and n_k
Initialize topic labels z_ij
for iterations in 1:n_iterations
  for each cell d_i
    for w_ij in d_i
      topic = z_ij
      peak = w_ij
```

$$n_{d, topic} = n_{d, topic} - 1$$
$$n_{peak, topic} = n_{peak, topic} - 1$$
$$n_{topic} = n_{topic} - 1$$
**for** k in topics
$$\quad p(z_{ij} = k \mid z^{(-i)}, w) \propto (n_{dk} + c_\alpha \alpha_k) \frac{n_{kw} + c_\beta \beta_w}{n_k + \sum_{w=1}^{V} c_\beta \beta_w}$$
**end**
Pick a new topic according to $p(z_{ij} = k \mid z^{(-i)}, w)$
$$n_{d, topic} = n_{d, topic} + 1$$
$$n_{peak, topic} = n_{peak, topic} + 1$$
$$n_{topic} = n_{topic} + 1$$
 **end**
 **end**
 **end**

In [4], the last Gibbs sampling iteration is chosen to be the output of the topic-peak and cell-topic matrices. We modify this approach slightly by calculating the posterior likelihood of each of the iterations and choosing the iteration with the highest likelihood. This is the maximum a posteriori (MAP) estimate.

## 1.2 Data transfer using a matrix prior

To achieve our goal of leveraging large scale data to improve inference in small datasets, we extend Algorithm 1 to accept a matrix prior $B$ to replace the vector prior $\beta$ for the topic-peak or topic-gene distribution. Unlike the vector prior $\beta$, which specifies the same prior distribution over vocabulary elements regardless of topic, the matrix prior $B$ can specify a different prior distribution for each topic. The matrix prior $B$ has elements $B_{tw}$ such that for each $t$, $\sum_w B_{tw} = 1$, corresponding to the probability of observing peak or gene $w$ in topic $t$. It has corresponding concentration parameter $c_B$. We modify the distribution of topics in Algorithm 1, $p(z_{ij} = k \mid z^{(-i)}, w)$, the probability of topic $z_{ij}$ being $k$ given all other topic assignments and all documents, in the algorithm by instead writing that

$$p(z_{ij} = k \mid z^{(-i)}, w) \propto (n_{dk} + c_\alpha \alpha_k) \frac{n_{kw} + c_B B_{kw}}{n_k + \sum_{w=1}^{V} c_B B_{kw}} \tag{1}$$

An interpretation of $c_B B_{tw}$ is that we are adding pseudocounts of the peak or gene $w$ to the topic $k$. This gives us finer control over the prior distribution of the topic-peak or topic-gene matrix. A key feature of using a matrix prior is that we can use the inferred $\phi$ from one LDA model as the prior for another model with the same vocabulary and similar topics, thereby transferring information from one model to another. We note that this proposed method is similar to a method first proposed in [9], but the exact form of the prior is different due to the hyperparameter $c_B$.

We call the dataset that we use to generate the matrix prior the "reference dataset" and the dataset to which we apply the prior the "target dataset." In order to transfer information from the reference to the target, we first run the LDA algorithm on the reference, where both $\alpha$ and $\beta$ are set to be the "uniform priors" described in Algorithm 1; in other words, $\alpha_i = 1/T$ and $\beta_j = 1/V$. This outputs an estimate of the topic-peak or topic-gene matrix, $\hat{\phi}_{ref}$, that contains information about the distribution of peaks or genes in each topic. We set the prior $B = \hat{\phi}_{ref}$, with a concentration parameter $c_B$. As the concentration parameter increases, the LDA algorithm upweights $\hat{\phi}_{ref}$.

## 2 Methods

### 2.1 Data

**2.1.1 Simulated data.** To evaluate the performance of the matrix prior methodology, we first simulated scATAC-seq data and compared inferred topics to the true generative topics that we used to create the synthetic data according to the LDA generative process (Section 1.1). We fixed the true topic-gene distribution across all simulated datasets, with peak or gene distributions for each topic simulated with a Dirichlet distribution with parameters $c_\beta = 0.1$, $\beta_j = 1/V$, resulting in $V$-element vectors. We simulated a new cell-topic matrix for each new dataset, also using a Dirichlet distribution, with parameters $c_\alpha = 0.3$, $\alpha_i = 1/T$ resulting in a $T$-element vector. We set the number of peaks/genes $V = 8000$, the number of topics $T = 30$, and the number of reads/UMIs per cell to be on average $\xi = 4000$.

In the *true matrix simulation*, the goal was to evaluate the matrix prior when the true topic-gene matrix was used as the matrix prior in LDA of simulted data. We simulated target data-sets with 1000, 2000, 4000, and 8000 cells, all with the same topic-gene matrix. For each dataset size, we simulated five datasets. In the *inferred matrix simulation*, instead of providing the true topic-gene matrix, we inferred the topic-gene matrix by performing an LDA of simulated reference data using a uniform symmetric prior, then provided that inferred matrix as the prior to LDA of simulated target data sets. We generated four synthetic reference datasets: one for each of 1000, 2000, 4000, or 8000 cells, and we simulated four target datasets of 1000 cells. We performed LDA with a uniform symmetric prior on each reference dataset, and used each resulting topic-gene output matrix as a prior for LDA of each target dataset.

**2.1.2 *C. elegans* scATAC-seq data.** Recently published *C. elegans* scATAC-seq data demonstrated increased resolution of cell types compared to bulk ATAC-seq data [1]. The cells were collected from animals in larval stage 2, and the authors used LDA followed by clustering to identify cell types, which are the labels we use here. The dataset has 30,764 cells and 13,734 peaks. We split the cells uniformly at random into a "reference dataset" of 27,764 cells and a "target dataset" of 3,000 cells to investigate the performance of the matrix prior.

**2.1.3 SHARE-seq mouse skin data.** SHARE-seq is a co-assay that generates both scA-TAC-seq and scRNA-seq data from the same single cells simultaneously [6]. For our analysis we selected the mouse skin data set from the original publication, which has 34,774 cells from 22 cell types. The scATAC-seq data for these cells consist of chromatin accessibility across 344,592 peaks, while the scRNA-seq data contain the expression measurements of 22,813 genes. First, we repeated our experiments from the *C. elegans* dataset, testing the effectiveness of the matrix prior for transferring information from a larger reference dataset to a smaller target dataset. Second, we leveraged the co-assay data, in which we know the ground-truth pairing of scATAC-seq and scRNA-seq measurements for each cell, to investigate whether we could use the matrix prior to transfer information between scRNA-seq and scATAC-seq datasets.

### 2.2 LDA analysis

**2.2.1 LDA analysis of simulated data.** For our analysis of simulated data, we used the implementation of the LDA algorithm that employs the Gibbs sampling scheme described in Algorithm 1 [1]. We set the number of Gibbs sampling iterations to be 1000. The iteration with the highest posterior probability was used to infer the cell-topic and topic-gene matrices.

In the true matrix simulation, we used the true generative topic-gene matrix as the matrix prior for the target data LDA. We compared the quality of the inferred cell-topic and topic-gene matrices both using the matrix prior and using the uniform prior. For the uniform prior,

we supplied LDA with the parameters $c_\alpha = 0.3$, and $c_\beta = 0.1$, matching the simulation Dirichlet parameters. When we used the matrix prior, we supplied LDA with the concentration parameter $c_\alpha = 0.3$, which matched the simulation Dirichlet parameter, and we set the matrix prior concentration parameter to $c_B = 1000$.

In the inferred matrix simulation, we used LDA with a uniform prior to infer a topic-gene matrix $\hat{\phi}$ using each of the four reference datasets with between 1000 and 8000 cells. We trained these uniform prior LDA models with $c_\alpha = 0.3$, and $c_\beta = 0.1$. We then used each resulting $\hat{\phi}$ as the matrix prior for each of the four target datasets. We again compared using the matrix prior to using a uniform prior. We used the same settings for the matrix prior LDA models as in the true matrix simulation.

**2.2.2 LDA analysis with experimental data: Transfer from scATAC-seq to scATAC-seq.** To assess whether using a matrix prior would improve LDA performance compared to a uniform prior on small, sparse data, we first trained an LDA model with a uniform prior on all available cells to compare to using our prior on a small dataset. We will refer to this model as the "joint model". Next, we split the data into a larger "reference set" of 31,774 cells and a smaller "target set" of 3000 cells, and trained a uniform prior LDA model on each of these sets. The output from the target data when the uniform prior was used set the floor for the expected performance of our prior. Last, we used the gene-topic probabilities from the reference model as a matrix prior for a new LDA model of the target set of cells, and we compared the results of this model with those of the uniform models to evaluate the effectiveness of the matrix prior.

**2.2.3 Hyperparameter search.** We conducted a hyperparameter search to inform our selection of values for the number of topics to use, $T$, and the two concentration parameters $c_\alpha$ and $c_B$. We used a grid-search strategy, testing seven values for $T$ (2, 3, 4, 5, 10, 15, and 20), four values for $c_\alpha$ (0.03, 0.3, 3.0, and 30.0), and eight values for $c_B$ (10, 50, 75, 150, 250, 1000, 2000, and 4000); and we employed a likelihood-based measure, perplexity, as our evaluation metric, following [10]. Perplexity is the negative exponent of the likelihood, and is calculated for each cell as $\exp\left(-\frac{\mathcal{L}(\mathbf{w}|\mathbf{z},\phi,c_\alpha,c_B)}{N}\right)$, where $N$ is the number of reads in a cell. A lower value of perplexity is better. To calculate the likelihood required for the perplexity measure, we use the Chib-style estimation procedure [11], which is a method using a Markov chain to evaluate $\mathcal{L}(\mathbf{w} \mid \mathbf{z}, \phi, c_\alpha, c_B)$.

Due to our two-tiered method, in which we train a uniform prior LDA on a reference subset of the data and then use the output as a matrix prior for an LDA on the target subset of the data, we also required two corresponding tiers for our hyperparameter search. Thus for the first tier, we generated a set of matrix priors by training, for each value of $T$ in our grid search, a uniform prior LDA model on the reference data set. All of these models used $c_\alpha = 3$ and $c_\beta = 800$, which we chose based on the hyperparameter values reported in [1]; note that we lowered $c_\beta$ compared to the published value of 2000 to allow the data to have a greater role in determining the topic-gene matrices we would use as matrix priors.

In the second tier of the hyperparameter search, we conducted the full grid search on the target data set with a range of values for $T$, $c_\alpha$, and $c_B$. We split the target data into ten different training/test splits of 2700 and 300 cells, and then trained each of the ten splits on each of the hyperparameter combinations, with the LDA using the matrix prior from tier 1 that corresponded to each value of $T$. Then, we evaluated the performance of each hyperparameter combination by computing the perplexity on the held out test set cells.

We found that the optimal number of topics was 10, that the perplexity was relatively insensitive to the choice of $c_\alpha$, and that perplexity dropped with increasing values of $c_B$. Following

[1], which suggested that LDA models were robust to extra topics, we increased the number of topics to add some flexibility to the model, and set $T$ to be 15 in our experiments. The results of the hyperparameter search and further comments are in S1 and S2 Figs. UMAP plots of the results of LDA for different numbers of topics are shown in S3 and S4 Figs.

We conducted the hyperparameter search for our LDA modeling of the SHARE-seq mouse skin data in similar fashion to the hyperparameter search for the *C. elegans* data. However, the SHARE-seq scATAC-seq dataset contains about ten times more reads per cell than the *C. elegans* dataset, leading to much longer LDA training times per cell and higher memory usage. To make the hyperparameter search more efficient, we limited the SHARE-seq analysis to the 20,000 peaks with the highest variance amongst cells, and randomly down-sampled the dataset to 7,000 cells. We split the 7000 cells into 6300 cells for the reference dataset, and 700 cells for the target dataset. We then made ten train/test splits of the target data set by sampling 630 training cells and using the remaining 70 held-out test cells for the Chib method. As with the *C. elegans* data, our hyperparameter search results show that per-plexity was not very sensitive to the $c_\alpha$ parameter, and that perplexity decreased as the value of the concentration parameter $c_B$ increased, suggesting that the matrix prior was able to improve the quality of the our inference (S2 Fig). However, surprisingly, our hyperparameter search achieved the lowest perplexity with just $T = 2$ topics. We conjectured that this may be because of the low number of peaks included in the analysis, the low number of cells, or the peaks with highest variance may not be able to distinguish the cells. It is also possible that without a normalization for the mean accessibility, we might not select for peaks with the most meaningful differences in accessibility. We hence opted to instead continue using 15 topics as in the *C. elegans* case. UMAP plots of the results of LDA for different numbers of topics are shown in S4 Fig, and we found that 10–15 topics visually had good separation of the previously called cell types.

**2.2.4 LDA analysis with experimental data: Transfer from scRNA-seq to scATAC-seq.** To use scRNA-seq data to generate a matrix prior for the LDA of the scATAC-seq data, our method requires that the scATAC-seq and scRNA-seq data share the same vocabulary. Hence, we translated the scATAC-seq data from a vocabulary of peaks to one of genes by counting the number of ATAC-seq cut sites per cell that overlapped each gene and its promoter (defined as the 2kb region immediately 5' of the transcription start site). We defined cut sites as reported in [1]; briefly, they are 60 bp regions centered on the mapping locations of the 5' ends of the paired end reads (which define the extent of the original DNA fragment that was cut out of the genome). We investigated whether this translated scATAC-seq data retained similar information to the native scATAC-seq data by qualitatively comparing UMAP plots of the LDA output of the raw peaks versus the summed cut sites and saw that the plots were qualitatively similar (S5 Fig). Furthermore, we investigated the similarity between the scRNA-seq data and the translated scATAC-seq data by testing the correlation between the two data sets based on the number of counts per cell and counts per gene (S6 Fig). We found that there was a moderate amount of correlation between the scRNA-seq data and the scATAC-seq data, suggesting that generating a matrix prior from scRNA-seq data and applying it to an LDA of a translated scA-TAC-seq data set may be useful. See S1 Note for results and discussion of our work in this direction.

## 2.3 Evaluation

**2.3.1 Evaluation of LDA in simulated data.** We used the mean squared error (MSE) to evaluate our inferred cell-topic matrix $\hat{\theta}$ and topic-gene matrix $\hat{\phi}$. Then, we calculated the

MSEs for the cell-topic matrix and topic-gene matrix using

$$\frac{1}{UT}\sum_{i=1}^{U}\sum_{j=1}^{T}(\hat{\theta}_{ij} - \theta_{ij})^2 \text{ and } \frac{1}{TV}\sum_{i=1}^{T}\sum_{j=1}^{V}(\hat{\phi}_{ij} - \phi_{ij})^2 \tag{2}$$

where $U$ is the number of cells, and $T$ is the number of topics.

In addition to MSE, we calculated Pearson's $r$ and Spearman's $r$ after flattening the cell-topic and topic-gene matrices.

One complication in simulation is that the order of topics inferred by LDA will not necessarily match the order of topics in the simulated ground truth; i.e. topic 1 from the output of LDA is not necessarily semantically the same as topic 1 in the simulated true cell-topic matrix. Therefore, prior to calculating any performance measure, we must match topics. We do this by using a greedy approach on the topic-peak or topic-gene matrix, considering each pair of topics in sorted order by Euclidean distance and allowing only one-to-one matches. For each topic, we match it to the true topic that is closest in MSE.

We used this greedy topic matching algorithm when evaluating the uniform prior LDA models in both the true matrix simulation and the inferred matrix simulation, and when evaluating the matrix prior LDA model for the inferred matrix simulation. We did not need to match topics for the matrix prior LDA in the true matrix simulation, because providing the true topic-gene distributions as the prior already imparts the proper topic semantics to the target LDA, making the inferred topic-gene matrix and the true topic-gene matrix directly comparable.

**2.3.2 Evaluation of LDA in experimental data.** Unlike our simulation experiments, in which we know the true underlying topic distributions, we do not have a ground truth to compare against for our analyses of datasets derived from biological experiments. Instead, to evaluate the performance of our LDA models, we compared the output of each LDA on a target subset to the LDA output when training on the full data set (i.e. the "joint model"). This has the interpretation that if our matrix prior allows the smaller target data set LDA to infer similar topics to the joint model, then our matrix prior has succeeded. The joint model results in topic assignments for all cells, inclusive of both cells assigned to be reference cells and those assigned to be target cells. Hence, to compare the joint model with the target LDA, we selected the rows in the cell-topic matrix of the joint model that corresponded to the target dataset cells. We then applied our greedy topic matching algorithm to account for any topic-switching that occurred in the LDA of the reference data set compared to the joint model. Finally, we evaluated the similarity between the outputs of the matrix prior LDA and the joint model using Pearson's correlation, Spearman's correlation, and MSE.

In addition to our quantitative evaluation of the matrix prior approach, we qualitatively evaluated its performance by visualizing the cell-topic matrix using UMAP [12]. For the *C. elegans* data, we took the cell-topic matrix LDA output from one of the splits of the target training set, computed a two dimensional UMAP embedding, and represented the embedding as a scatter plot in which we colored the cells based on their published cell type labels [1]. We quantitatively measured how well the inferred cell-topic matrices could separate cells of different types (both at the level of broader cell types and more specific ones, such as neuron subtypes) by using the silhouette coefficient. The silhouette coefficient is a distance-based measure of cluster cohesiveness that has previously been used as a performance metric when comparing the performance of different single cell analysis methods [13], and it is computed using Eq 3. For a cell with index $i$, and $C_i$ the set of indices of cells with the same cell type label as cell $i$, $a(i) = \frac{1}{C_i - 1}\sum_{j \in C_i, i \neq j} d(i,j)$ is the average Euclidean distance to all other cells of the same cell type. We similarly define $b(i) = \min_{k \neq i} \frac{1}{C_k}\sum_{j \in C_k} d(i,j)$ to be the lowest average Euclidean

distance to a different cell type from cell $i$.

$$\frac{b(i) - a(i)}{\max(b(i), a(i))} \tag{3}$$

The resulting silhouette coefficent is bounded between -1 and 1. Values close to 1 indicate that a point is closer to points in the same cluster than to points in other clusters, and a high silhouette coefficient indicates that the LDA topics reflect the published cell types well.

## 3 Results

### 3.1 Matrix prior improves topic inference in simulation

We began testing the matrix prior by running LDA on simulated data derived from known cell-topic and topic-gene matrices. In the true matrix simulation, we tested the hypothesis that an LDA given the true topic-gene matrix as a prior would yield results more similar (by MSE) to the ground truth than the uniform prior LDA would. We also tested the hypothesis that increasing the number of target cells would reduce the performance advantage of the matrix prior over the uniform prior. In the inferred matrix simulation, we tried using LDA output on a reference data set as our matrix prior for the target LDA and tested the effect of varying the reference data set size on the performance of the target LDA.

In the true matrix simulation, we found that using the matrix prior instead of the uniform prior led to more accurate inference of the topic-gene and cell-topic matrices (Fig 1a). We kept the weight of the matrix prior, $c_B$, constant and varied the number of cells in the target dataset to understand effects of target dataset size. The MSE was higher when we used a uniform prior than when we used the matrix prior regardless of target dataset size (we tested 1000, 2000, 4000, and 8000 target cells), although the difference in performance decreased as the cell number increased, suggesting that with more cells the data began to overwhelm the prior. To better understand how the matrix prior improves the LDA results, we analyzed some representative uniform prior LDA results in more detail. When the number of cells was low, the uniform prior LDA underestimated the weights of high probability topics, and even at 8000 cells, it incorrectly predicted topic assignments in some cells (S7 Fig, top row). On the other hand, for the matrix prior LDA, the inferred and true cell-topic matrices agreed (S7 Fig, bottom row). We found similar results for the topic-gene matrices (S8 Fig).

In the inferred matrix simulation, we found that the matrix prior improved the inference of the cell-topic and the topic-gene matrices compared to the uniform prior, as measured by MSE against the ground truth (Fig 1a and 1b). We also trained a series of LDA models on a 1000 cell target data set, each with a matrix prior generated from a uniform LDA trained on a reference dataset with 1000, 2000, 4000, or 8000 cells. As the number of reference cells increased, the MSE continually improved (x-axis), and we note that even the matrix prior generated from a reference dataset of only 1000 cells improved LDA performance compared to a uniform prior (Fig 1c and 1d). In representative simulated datasets, the cell-topic and topic-gene matrices more closely followed the $y = x$ line as the number of reference cells increased (S9 Fig). These two simulations suggest that under ideal conditions, the matrix prior method is able to improve the quality of inferred topics in LDA.

### 3.2 Whole worm scATAC-seq prior improves concordance with joint model

We used the *C. elegans* data to validate the ability of our matrix prior to improve LDA inference on real data. We randomly split the 34,764 cells into a 3,000 cell target dataset and a

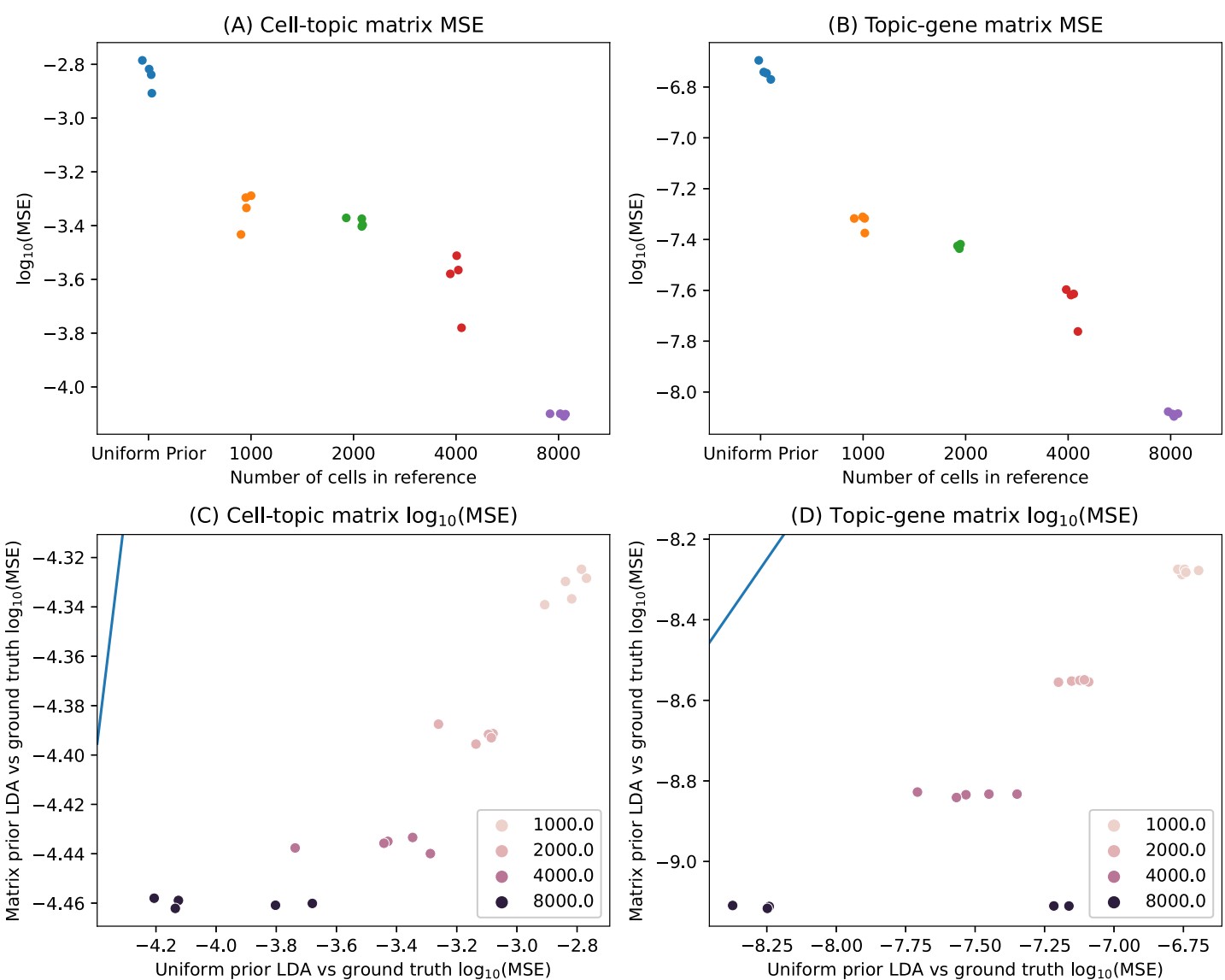

**Fig 1. Simulation experiments show that the matrix prior improves the concordance between inferred topics and the ground truth compared to the uniform prior.** Experiments from the true matrix simulation and the inferred matrix simulation are shown here for different numbers of cells in the target dataset (different colors). A: MSE from the ground truth to the LDA with a ground truth matrix prior (y-axis) is plotted against the MSE from the ground truth to the uniform symmetric prior LDA (x-axis), for both the cell-topic matrix and B: the topic-gene matrix. Each point represents one independently simulated dataset, with a unique true cell-topic matrix and topic-gene matrix. C: MSE to the ground truth for the LDA with a matrix prior inferred from a simulated reference dataset is shown for different reference data set sizes. MSE is plotted for both the cell-topic matrices and D: the topic-gene matrices. The blue line is the line $y = x$.

27,764 cell reference dataset, and used the split data to train a matrix prior LDA on the target dataset. Then, we trained a separate uniform prior LDA on the full *C. elegans* dataset (the "joint model"), and compared the inferred topics between the matrix prior LDA and the joint model. We also trained a uniform prior LDA on the target dataset, and evaluated whether the matrix prior LDA results were more similar to the the joint model than the uniform prior LDA results.

We found that the Pearson correlation between the target LDA and the joint model was higher when we used the matrix prior than when we used the uniform prior (Fig 2, left). The correlation increased with increasing $c_B$, but eventually reached a saturation point. We also

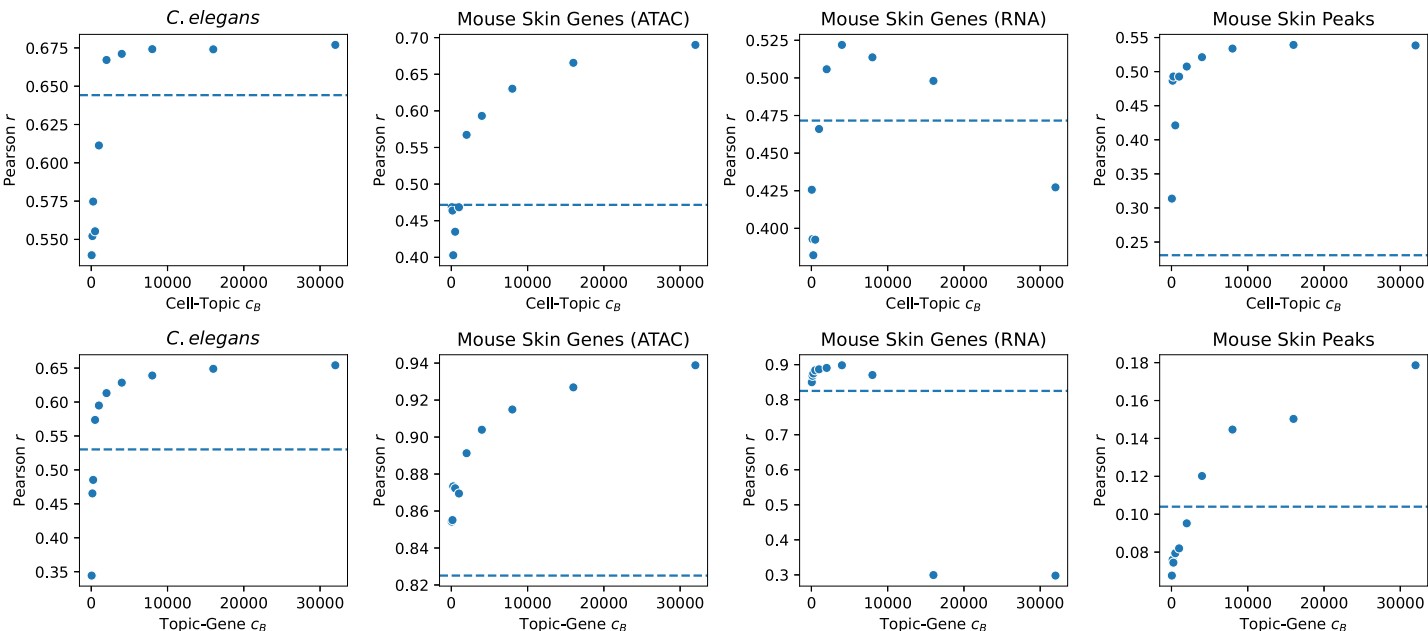

**Fig 2. Correlation of the target set with the matrix prior versus the joint model.** The Pearson correlation between LDA results on the target dataset for the matrix prior LDA and the full dataset uniform symmetric LDA (the "joint model") increases as $c_B$ increases. Pearson $r$ values are plotted as a function of $c_B$ for the cell-topic (top row) and topic-gene (bottom row) matrices for LDA experiments on four different datasets: *C.elegans* scATAC-seq data (first column), SHARE-seq mouse skin scATAC-seq data with the peak vocabulary translated to genes (second column), SHARE-seq mouse skin scRNA-seq data (third column), and SHARE-seq mouse skin scATAC-seq data using the peak vocabulary (fourth column). The dotted horizontal lines indicate the correlation between the uniform prior LDA and the joint model.

found that the matrix prior outperformed the uniform prior in Spearman correlation and MSE (S10 Fig). Other measures are also reported in S10–S13 Figs. When plotting the values of the cell-topic and topic-gene matrices, the points near the $y = x$ line for the cell-topic matrix tightened as the concentration parameter $c_B$ increased (S14 Fig). We also additionally note that in our hyperparameter search, we found evidence that the use of the matrix prior improved the quality of the topic-gene matrix, since the perplexity value in the held out test set improved as the matrix prior concentration parameter increased (S1 Fig).

### 3.3 SHARE-seq scATAC-seq matrix prior improves concordance with the joint model

We used data from mouse skin cells analyzed using the SHARE-seq assay [6] to further validate the ability of a matrix prior to improve inference in LDA. We split the 34,774 cells into a 31,774 cell reference dataset and a 3000 cell target dataset, and the same analyses were applied as in the *C. elegans* data (Section 3.2).

We note that increasing the value of $c_B$ more consistently improved the correspondence between the target LDA and the joint model for the topic-gene matrix than the cell-topic matrix. This difference is most likely due to the fact that the matrix prior is specified as a prior on the topic-gene distribution, and thus only influences the cell-topic distribution indirectly through the training of the LDA model.

In addition to serving as another dataset to validate our scATAC-seq prior, the SHARE-seq co-assay data allowed us to evaluate whether a matrix prior generated from one data modality, scRNA-seq, could improve LDA performance on another data modality, scATAC-seq. Because the SHARE-seq scATAC-seq and scRNA-seq data were generated from the same cells, we were able to directly assess the agreement between the scRNA-seq LDA and the scATAC-seq

LDA, with and without a matrix prior derived from the scRNA-seq data (Section 2.2.4). See S1 Note, where we report that the scRNA-seq prior was able to improve inference for moderate values of $c_B$ but worsened inference for larger values. Note that although scRNA-seq and scATAC-seq produce data on different scales, this does not affect the matrix prior because it is based on the topic-gene probabilities (which sum to 1 for each topic) and not the counts.

### 3.4 Mouse skin cell types are more clearly separated with the use of the matrix prior

We next aimed to assess whether using a matrix prior would improve the ability of LDA to distinguish the cell types in a small subset of the SHARE-seq mouse skin data set [6]. We split the data into a reference data set and a target data set (see Methods), and then we analyzed the target data set using both a uniform prior and a matrix prior derived from LDA on the reference data set. After applying UMAP to our two models to reduce the 15-dimensional topic space into a two-dimensional UMAP space, we qualitatively observed that the matrix prior LDA resulted in cell clusters that better agreed with the published cell type labels than the uniform prior LDA, and this improvement became more marked as we increased the weight of the prior, $c_B$ (Fig 3). We used the silhouette score to quantitatively measure how well the cells

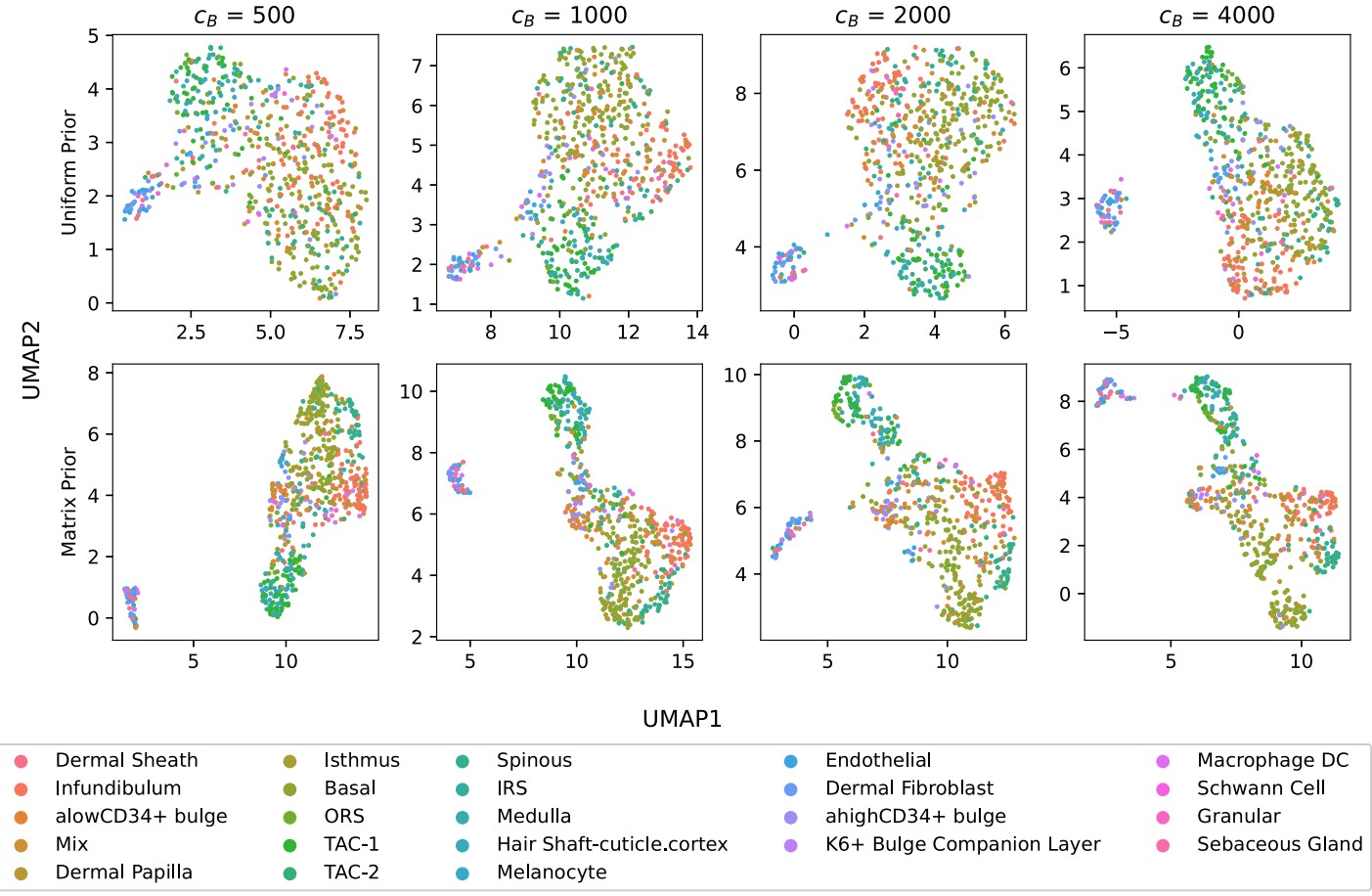

**Fig 3. Qualitative improvements when using the matrix prior.** Increasing the weight of the matrix prior (bottom row) shows a qualitative improvement in the ability of the target dataset LDA to discriminate among cell types compared the uniform prior (top row). UMAP embeddings of the cell-topic matrices from SHARE-seq mouse skin scATAC-seq data using the peak vocabulary are trained with different values of $c_B$ (different columns). Scatter points representing cells are colored by their published cell type annotations.

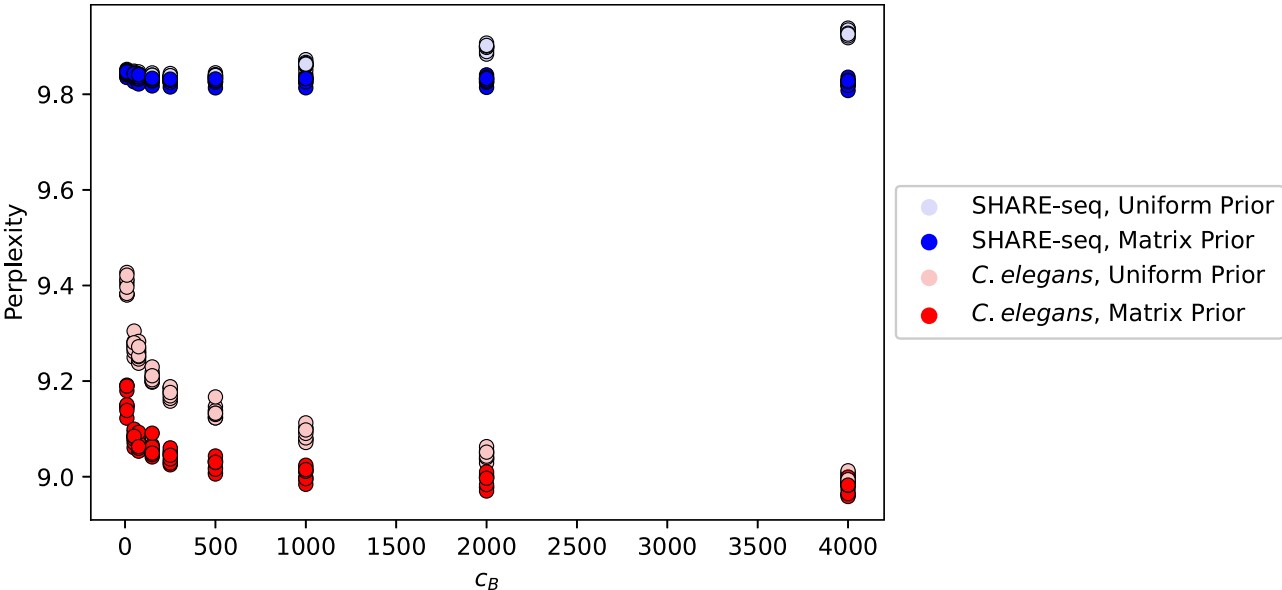

**Fig 4. Quantitative improvement of perplexity values with the matrix prior.** Perplexity values (y-axis) demonstrate quantitative improvement of the LDA model after using the matrix prior (darker colors) compared to the uniform prior (lighter colors) for various values of the weight of the prior (x-axis). The same procedure was used for both the SHARE-seq data set (blue) and the *C. elegans* data set (red). Each point is a separate split of the target data into a test and a training set.

clustered by their cell type labels (S16 Fig), but the silhouette values did not improve with use of the matrix prior. We conducted a similar experiment in *C. elegans data* (S2 Note), but did not see qualitative improvement of cell clusters in *C. elegans*. A possible reason for this is that even in the case of the uniform prior, LDA created separation in the *C. elegans* cell types, and hence no further improvement was possible by using the matrix prior.

We also used the perplexity measure to quantitatively measure how well the model was doing when different weights of the prior were used (Fig 4). This is a method that is similar to that of the hyperparameter search (Section 2.2.3). We found that as we trained matrix prior models with increasing values of $c_B$, the perplexity decreased. This was true both for *C. elegans* and SHARE-seq. For both cases, the uniform prior resulted in greater (worse) values of perplexity. For *C. elegans*, perplexity values for both the uniform prior and matrix prior decreased as $c_B$ increased. On the other hand, for the SHARE-seq data, the perplexity values decreased for the matrix prior, but not for the uniform prior.

## 4 Discussion

We have shown through both simulation study and through analysis of real data that the matrix prior we propose is able to capture information from a larger reference dataset and impart the semantics of the topic-gene or topic-peak matrix onto a smaller target dataset. In our simulation studies, we found that when the true topic-gene and cell-topic matrices were known, we were able to recover those matrices both by directly inputting the truth as the prior and by inferring the topic-gene matrix from a reference data set (Fig 1). These simulations demonstrated that in ideal conditions, the matrix prior can greatly improve performance of LDA. In our real data examples, we examined *C. elegans* and mouse skin cell data. We saw promising results when we used scATAC-seq data to generate a prior for analyzing a target scATAC-seq dataset. We found that LDA results on a small target dataset were more concordant with a model of the full data set when we used a matrix prior derived from a larger

reference dataset than when we used a uniform prior (Fig 2). Furthermore, in the case of the mouse skin data, we showed qualitative improvements in cell type discrimination for the matrix prior LDA compared to the uniform prior LDA (S16 Fig).

We also attempted to transfer information across single cell data modalities by deriving a matrix prior from scRNA-seq data and applying it to a target scATAC-seq dataset. We found that with moderate values of the concentration parameter, the agreement between the outputs of LDA based on the target dataset and LDA based on the full data set improved when using the matrix prior compared to the uniform prior (S12 Fig). Leveraging information from multiple single cell data modalities is an area of active research. Some popular tools, like Seurat [14], scVI Tools [15], or LIGER [16], take multiple data sets as input and use embedding techniques to analyze them jointly. These approaches are powerful, but require manipulating potentially very large datasets every time one wants to add new data into the model. In contrast, our matrix prior approach requires just a single large upfront compute task to train an LDA on a reference dataset, which yields a compact gene-topic matrix that can be used as a matrix prior for training comparatively lightweight LDA models in all subsequent analyses of new datasets. We also anticipate that new approaches, such as Polarbear [17] and BABEL [18], that use deep learning models to translate data from one single cell modality to another will improve our ability to generate cross-modality matrix priors, not only between scATAC-seq and scRNA-seq data, but also between other pairs of modalities.

## Supporting information

**S1 Table. Glossary of variables used.**
(PDF)

**S1 Note. SHARE-seq scRNA-seq matrix prior may have limited use.** We investigated the use of the scRNA-seq matrix prior.
(PDF)

**S2 Note. *C. elegans* silhouette values.** We evaluated silhouette values in the *C. elegans* dataset.
(PDF)

**S1 Fig. Hyperparameter search for *C. elegans*.** Hyperparameter search for *C. elegans* data was used to optimize the parameters. Each point is the average of 10 folds of the perplexity value of the test set. The x-axis is the value of the hyperparameter, and the y-axis is the perplexity value.
(PDF)

**S2 Fig. Hyperparameter search for SHARE-seq data.** Hyperparameter search for SHARE-seq data with peaks was used to optimize the parameters. Each point is the average of 10 folds of the perplexity value of the test set. The x-axis is the value of the hyperparameter, and the y-axis is the perplexity value.
(PDF)

**S3 Fig. UMAP embeddings of *C. elegans* data using different numbers of topics.** UMAP embeddings for different numbers of topics are shown to provide intuition on the effect of the number of topics. Embeddings were made for cell-topic matrices from matrix prior LDA on *C. elegans* scATAC-seq data using a fixed value of $c_B = 4,000$ and 13,734 peaks, while varying the number of topics.
(PDF)

**S4 Fig. UMAP embeddings of SHARE-seq data using different numbers of topics.** UMAP embeddings for different numbers of topics are shown to provide intuition on the effect of the

number of topics. Plots were made with 630 target cells from the mouse skin SHARE-seq peak data subsetted to 7,000 cells and 20,000 most variable peaks. LDA was run with $c_\beta = 4,000$ using the uniform prior, while varying the number of topics.
(PDF)

**S5 Fig. UMAP of peaks versus cut sites summed over genes.** UMAP was applied to the cell-topic matrices output from LDA joint model to qualitatively compare cut sites summed over genes versus peaks. LDA was run with 15 topics, $c_\alpha = 3$, and $c_\beta = 4000$. On the left, the raw data fed into LDA are the cut sites summed over the 22,813 genes, as described in Section 2.2.4. On the right, the data fed into LDA are the 344,592 raw peaks.
(PNG)

**S6 Fig. Shared information between scATAC-seq and scRNA-seq.** RNA reads and scATAC-seq cut sites summed over gene bodies are compared to determine the shared information between the data modalities. On the left, the signal per cell is the log plus one of the total number of counts for each cell (i.e. summing across all the genes in a cell). On the right, the signal per gene is the log plus one of the total number of counts for each gene (i.e. summing across all the cells for a gene). The Pearson correlation is reported in each plot. A kernel density estimator is overlayed on the data. The x-axis shows the score for the scATAC-seq data, and the y-axis shows the score for the scRNA-seq data.
(PNG)

**S7 Fig. Cell-topic matrix true matrix simulation results.** Scatter plots of cell-topic matrix values demonstrate the improvement of the matrix prior over the uniform prior in the true matrix simulation and further show that the performance of the uniform prior approaches that of the matrix prior as the number of cells increases. Plots show simulated true values (x-axis) of the cell-topic matrix against inferred values using LDA (y-axis). Pearson $r$ (r) and Spearman $r$ (sr) are reported for each plot. We compared different numbers of cells in the target dataset (different columns). We compared LDA with a uniform prior (top row) with a matrix prior generated from the true topic-gene matrix (bottom row). The blue dotted line is the line $y = x$.
(PNG)

**S8 Fig. Topic-gene matrix true matrix simulation results.** Scatter plots of topic-gene matrix values demonstrate the improvement of the matrix prior over the uniform prior in the true matrix simulation and further show that the performance of the uniform prior approaches that of the matrix prior as the number of cells increases. Plots show simulated true values (x-axis) of the topic-gene matrix against inferred values using LDA (y-axis). Pearson $r$ (r) and Spearman $r$ (sr) are reported for each plot. We compared different numbers of cells in the target dataset (different columns). We compared LDA with a uniform prior (top row) with a matrix prior generated from the true topic-gene matrix (bottom row). The blue dotted line is the line $y = x$.
(PNG)

**S9 Fig. Inferred matrix simulation results.** Scatter plots demonstrate the improvement of the matrix prior in both the cell-topic and topic-gene matrices as the number of reference cells increases in the inferred matrix simulation. 1000 simulated cells were analyzed using a uniform prior (left-most column) and a matrix prior. The dotted red line is the $y = x$ line. True simulated values (x-axis) and inferred values (y-axis) are plotted for both the topic-gene matrices (top) cell-topic matrices (bottom).
(PNG)

**S10 Fig. Summary statistics from joint model analysis in *C. elegans*.** Summary statistics (y-axis) between output matrices of the joint model and LDA with matrix prior, both trained on the *C. elegans* data, show that as the weight of the prior increases, agreement between the matrix prior and joint model also increases. Each plot shows the matrix prior LDA results (points) for increasing values of $c_B$ (x-axis) versus the uniform prior (blue dotted line). The top row of plots shows summary statistics for the cell-topic matrix, and the bottom row of plots shows summary statistics for the topic-gene matrix.
(PDF)

**S11 Fig. Summary statistics from joint model analysis in SHARE-seq cut site data.** Summary statistics (y-axis) between output matrices of the joint model and LDA with matrix prior, both trained on the SHARE-seq mouse skin scATAC-seq data with cut sites summed over genes (i.e. using the genes vocabulary), show that as the weight of the prior increases, agreement between the matrix prior and joint model also increases. Each plot shows the matrix prior LDA results (points) for increasing values of $c_B$ (x-axis) versus the uniform prior (blue dotted line). The top row of plots shows summary statistics for the cell-topic matrix, and the bottom row of plots shows summary statistics for the topic-gene matrix.
(PDF)

**S12 Fig. Summary statistics from joint model analysis with SHARE-seq scRNA-seq data.** Summary statistics (y-axis) between output matrices of the joint model and LDA with matrix prior, both trained on the SHARE-seq mouse skin scRNA-seq data, show that as the weight of the prior increases, agreement between the matrix prior and joint model also increases with moderate weights and declines with higher weights. Each plot shows the matrix prior LDA results (points) for increasing values of $c_B$ (x-axis) versus the uniform prior (blue dotted line). The top row of plots shows summary statistics for the cell-topic matrix, and the bottom row of plots shows summary statistics for the topic-gene matrix.
(PDF)

**S13 Fig. Summary statistics from joint model analysis in SHARE-seq peak data.** Summary statistics (y-axis) between output matrices of the joint model and LDA with matrix prior, both trained on the SHARE-seq mouse skin scATAC-seq data (i.e. using the peaks vocabulary), show that as the weight of the prior increases, agreement between the matrix prior and joint model also increases. Each plot shows the matrix prior LDA results (points) for increasing values of $c_B$ (x-axis) versus the uniform prior (blue dotted line). The top row of plots shows summary statistics for the cell-topic matrix, and the bottom row of plots shows summary statistics for the topic-peak matrix.
(PDF)

**S14 Fig. Scatter plots for cell-topic matrices in joint analysis.** Comparing the cell-topic matrices of the joint model versus the LDA with the matrix prior reveals that as the weight of the matrix prior increases, the agreement between the models increases. The effect of different values of $c_B$ were evaluated by comparing the cell-topic matrix using the matrix prior to the cell-topic matrix from the joint model. Different values of $c_B$ are plotted across different columns, and different datasets are shown in different rows. We first flatten the cell-topic matrices so that they can be plotted. The cell-topic assignments from the joint model are shown on the x-axis, and the inferred cell-topic assignments from the matrix prior LDA are shown on the y-axis. A dotted red line is drawn to indicate the line $y = x$. Zero values are omitted from the plots, but the number of zeros exclusively in the cell-topic matrix of the joint model,

exclusively in the cell-topic matrix of the LDA with matrix prior, and number of zeros in both is noted below each plot.
(PDF)

**S15 Fig. Scatter plots for topic-gene matrices in joint analysis.** Comparing the topic-gene matrices of the joint model versus the LDA with the matrix prior reveals that as the weight of the prior increases, the agreement between the models increases. The effect of different values of $c_B$ were evaluated by comparing the topic-gene matrix using the matrix prior to the topic-gene matrix from the joint model. Different values of $c_B$ are plotted across different columns, and different datasets are shown in different rows. We first flatten the topic-gene matrices so that they can be plotted. The topic-gene assignments from the joint model are shown on the x-axis, and the inferred topic-gene assignments from the matrix prior LDA are shown on the y-axis. A dotted red line is drawn to indicate the line $y = x$. Zero values are omitted from the plots, but the number of zeros exclusively in the topic-gene matrix of the joint model, exclusively in the topic-gene matrix of the LDA with matrix prior, and number of zeros in both is noted below each plot.
(PDF)

**S16 Fig. Silhouette values as the concentration parameter changes.** The matrix prior and uniform prior show similar performance as a function of $c_B$. Average silhouette values for the published cell type annotations (y-axis) are plotted against increasing values of $c_B$ (x-axis). Different colored lines indicate whether the SHARE-seq data set (red) or the *C. elegans* data set (blue) was used. Each data set was analyzed using a uniform prior (lighter colors) and the matrix prior (darker colors).
(PDF)

**S17 Fig. Silhouette and UMAP of *C. elegans*.** A: UMAP plot of all the *C. elegans* data reveal cell type structure from LDA analysis with different weights $c_B$ of the matrix prior. Cells are colored based on their published cell type annotations. B: Silhouette plots demonstrate that in *C. elegans*, the silhouette value did not improve with increased weight of the prior. Silhouette values are shown for *C. elegans* cell types plotted for results from a uniform prior LDA model and matrix prior LDA models trained with increasing values of $c_B$ using 15 topics and the scATAC-seq data translated into the genes vocabulary (ATAC cut sites summed over the promoter and gene body for 13,734 genes). Each row in each plot represents one cell, and the silhouette value of the cell is the length of the line. The mean silhouette value for all of the cells is shown as "Overall", and the mean silhouette value for only the neurons is shown as "Neuron."
(PDF)

**S18 Fig. Silhouette and UMAP of common *C. elegans* celltypes.** A: A subset of S17 Fig (a) that includes only neurons, demonstrating that increased values of $c_B$ have little effect on the ability of the matrix prior LDA to distinguish among published cell types. Cells are colored by published neuron subtype labels. B: Silhouette plots of the neurons in the *C. elegans* dataset. The overall mean silhouette values and the mean positive silhouette values are reported.
(PDF)

## Author Contributions

**Conceptualization:** Timothy Durham, Louis Gevirtzman, William Stafford Noble.

**Data curation:** Alan Min, Timothy Durham.

**Formal analysis:** Alan Min.

**Investigation:** Alan Min, Timothy Durham.

**Methodology:** Alan Min, Timothy Durham.

**Project administration:** Timothy Durham, William Stafford Noble.

**Resources:** William Stafford Noble.

**Software:** Alan Min, Louis Gevirtzman.

**Supervision:** Timothy Durham, William Stafford Noble.

**Validation:** Alan Min.

**Visualization:** Alan Min.

**Writing – original draft:** Alan Min.

**Writing – review & editing:** Alan Min, Timothy Durham, William Stafford Noble.

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
