## [Decision Letter · Decision Letter 0]

12 Dec 2022

Dear Prof. Noble,

Thank you very much for submitting your manuscript "Matrix prior for data transfer between single cell data types in latent Dirichlet allocation" for consideration at PLOS Computational Biology.

As with all papers reviewed by the journal, your manuscript was reviewed by members of the editorial board and by several independent reviewers. In light of the reviews (below this email), we would like to invite the resubmission of a significantly-revised version that takes into account the reviewers' comments.

We cannot make any decision about publication until we have seen the revised manuscript and your response to the reviewers' comments. Your revised manuscript is also likely to be sent to reviewers for further evaluation.

Sincerely,

Qing Nie

Academic Editor

PLOS Computational Biology

Jian Ma

Section Editor

PLOS Computational Biology

Reviewer's Responses to Questions

**Comments to the Authors:**

**Reviewer #1: **The authors developed a latent Dirchlet allocation-based method, which could leverage information from previously generated large scale single cell genomics data to guide the analysis of new single cell data. The approach is useful for the new data annotation. The authors illustrated the method’s ability on both simulation and real datasets. Below are the specific comments:

1. Among all the cases analyzed, the cell types in the reference sets and target sets are the same. However, there might be some target set-specific cell types in the newly generated single cell data, especially, the data generated with different conditions such as control and treatment experiments. How is the method’s performance on such data? Could this method capture target set-specific cell types with new topic? This is very important when applying to new data.

2. In section 3.3.2, the procedure to make the comparison between the matrix prior and the joint model was not clearly described. Could you please make it clearer?

3. How is the number of topics K selected?

4. In section 4.3, the authors implemented the method on SHARE-seq data, which has both scRNA-seq and scATAC-seq from the same cell. How did the authors deal with the different scale between scRNA-seq and scATAC-seq data?

**Reviewer #2: **In this manuscript, the authors Min et al. developed a novel approach that leverage the information from the published large-scale single-cell genomic data to facilitate the analysis of new datasets. The approach could be very helpful for biologists working on single-cell epigenomic analysis, as it has been challenging to generate large-scale datasets due to the high cost of library preparation and sequencing. I am generally satisfied with the performance of the method and excited about its potential applications. Following are several comments that should be fixed before publication.

The authors should evaluate the performance of the technique for leveraging information from different datasets (e.g., generated by different scATAC-seq techniques or in different experiment batches).

How does the data sparsity of scATAC-seq affect the performance of the method? It would be interesting to check whether the approach can be used to improve the clustering analysis of data with relatively shallow sequencing.

To facilitate the application of the method, it would be great if the authors could include an example (e.g., a test dataset, processing pipeline and analysis result) in the GitHub.

**Have the authors made all data and (if applicable) computational code underlying the findings in their manuscript fully available?**

Reviewer #1: Yes

Reviewer #2: Yes

PLOS authors have the option to publish the peer review history of their article (what does this mean?). If published, this will include your full peer review and any attached files.

Reviewer #1: No

Reviewer #2: **Yes: **Junyue Cao
---

## [Decision Letter · Decision Letter 1]

26 Mar 2023

Dear Prof. Noble,

We are pleased to inform you that your manuscript 'Matrix prior for data transfer between single cell data types in latent Dirichlet allocation' has been provisionally accepted for publication in PLOS Computational Biology.

Best regards,

Qing Nie

Academic Editor

PLOS Computational Biology

Jian Ma

Section Editor

PLOS Computational Biology

Reviewer's Responses to Questions

**Comments to the Authors:**

Reviewer #1: The authors have addressed my concerns.

Reviewer #2: The authors have addressed the key concerns of my comments. The code (https://github.com/Noble-Lab/lda) should be made available before the paper is accepted for publication.

**Have the authors made all data and (if applicable) computational code underlying the findings in their manuscript fully available?**

Reviewer #1: Yes

Reviewer #2: **No: **https://github.com/Noble-Lab/lda is not available for now.

PLOS authors have the option to publish the peer review history of their article (what does this mean?). If published, this will include your full peer review and any attached files.

Reviewer #1: No

Reviewer #2: **Yes: **Junyue Cao

---

## [Editor Report · Acceptance letter]

2 May 2023

PCOMPBIOL-D-22-01702R1 

Matrix prior for data transfer between single cell data types in latent Dirichlet allocation

Dear Dr Noble,

I am pleased to inform you that your manuscript has been formally accepted for publication in PLOS Computational Biology. Your manuscript is now with our production department and you will be notified of the publication date in due course.

With kind regards,

Zsofi Zombor
